# Spell4TTS: Acoustically-informed spellings for improving text-to-speech pronunciations

*Jason Fong, Hao Tang, Simon King*

The Centre for Speech Technology Research, University of Edinburgh, UK

{jason.fong, hao.tang, simon.king}@ed.ac.uk

## Abstract

Ensuring accurate pronunciation is critical for high-quality text-to-speech (TTS). This typically requires a phoneme-based pronunciation dictionary, which is labour-intensive and costly to create. Previous work has suggested using graphemes instead of phonemes, but the inevitable pronunciation errors that occur cannot be fixed, since there is no longer a pronunciation dictionary. As an alternative, speech-based self-supervised learning (SSL) models have been proposed for pronunciation control, but these models are computationally expensive to train, produce representations that are not easily interpretable, and capture unwanted non-phonemic information. To address these limitations, we propose Spell4TTS, a novel method that generates acoustically-informed word spellings. Spellings are both interpretable and easily edited. The method could be applied to any existing pre-built TTS system. Our experiments show that the method creates word spellings that lead to fewer TTS pronunciation errors than the original spellings, or an Automatic Speech Recognition baseline. Additionally, we observe that pronunciation can be further enhanced by ranking candidates in the space of SSL speech representations, and by incorporating Human-in-the-Loop screening over the top-ranked spellings devised by our method. By working with spellings of words (composed of characters), the method lowers the entry barrier for TTS system development for languages with limited pronunciation resources. It should reduce the time and cost involved in creating and maintaining pronunciation dictionaries.

**Index Terms**: speech synthesis, grapheme-input, pronunciation control

## 1. Introduction

Text-to-speech (TTS) has an ever-expanding, diverse range of applications – virtual assistants, audiobooks, navigation systems, and many more – but they all need accurate pronunciations. The majority of TTS systems therefore use phonemes as an intermediate representation between the text and the acoustic model. Predicting phonemes from the input text requires a pronunciation lexicon and/or a grapheme-to-phoneme method (whether rules or a trained model), both of which are costly and time-consuming to create. In contrast, using graphemes as input to the acoustic model eliminates the need for these resources [1, 2] and thus offers a promising solution for scaling TTS to many more of the world's languages.

However, accurate pronunciation remains a challenge due to the complexity and ambiguity in mapping graphemes to speech sounds [3]. Consequently, the implicit pronunciation prediction performed within the front-end of neural grapheme-based TTS models will be error-prone [4, 5, 6], especially for words not seen in the training data. Worse, since it is performed *implicitly*, there is no obvious way to correct these errors *within the model*.

### 1.1. Prior work and alternative solutions

Although it is theoretically possible to obtain accurate pronunciations by training grapheme-based TTS on a dataset that covers all possible word types, this approach is impractical. Furthermore, because new words are constantly being created, it would be very difficult to keep such a dataset up-to-date.

To address the challenge of mispronunciations in grapheme-based TTS, one potential solution involves requesting annotators to transcribe grapheme sequences that the TTS model then accurately pronounces. However, this trial-and-error approach is time-consuming and annotators must become familiar with how the TTS model pronounces its inputs. Furthermore, since the TTS model's pronunciation of graphemes may change with each retraining, this approach necessitates re-transcription, making it even more costly.

Another alternative solution is to use a speech-based lexicon to control TTS pronunciations [7]. Speech-based lexicons have demonstrated promising results in correcting pronunciations, and are more affordable to commission than phoneme-based ones. However, this solution relies on large pretrained self-supervised speech representation models such as wav2vec 2.0 [8] or HuBERT [9], which are computationally expensive to train and require large amounts of speech data. This limitation reduces its usefulness for low-resource languages. Additionally, self-supervised speech representations are less interpretable compared to graphemes and still retain prosodic and speaker-specific information [10], which may reduce their effectiveness as reliable stand-ins for pronunciation [11, 7].

### 1.2. Proposed method

Our proposed method circumvents these limitations by finding a sequence of graphemes for a word that – when input to a grapheme-based TTS model – results in a pronunciation close to a ground-truth spoken example. These sequences may be stored in a simple dictionary, to be used during synthesis instead of that word's original spelling. Our method is a three-stage process. First, given a time-aligned spoken example for a word, an Automatic Speech Recognition (ASR) system generates multiple candidate spellings (e.g., hundreds or thousands). Second, a pre-existing grapheme-based TTS system synthesises all these candidates. Third, the best spelling is chosen, based on acoustic distance between synthetic and ground-truth speech. We provide experimental results for a variety of acoustic distances, including a novel one using self-supervised (SSL) speech representations.

The output of the method is a spelling that results in a

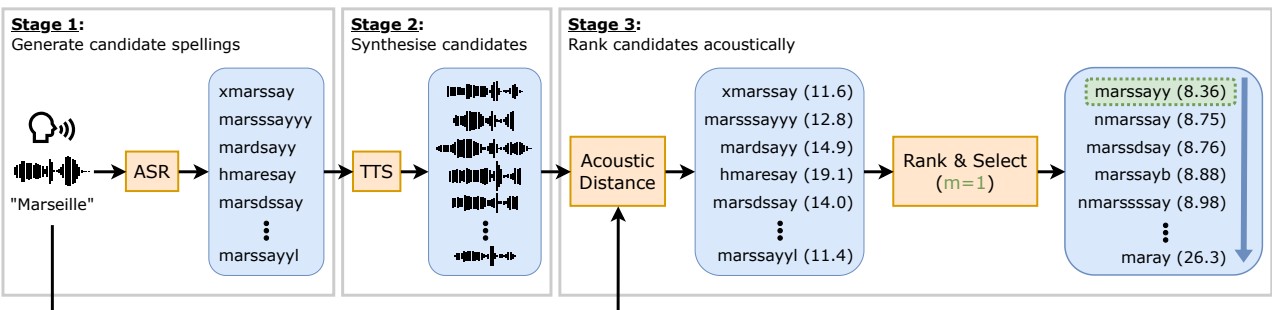

Figure 1: *Overview of Spell4TTS, our proposed method for automatically finding word spellings from spoken examples, for improving TTS pronunciation. Pictured are actual candidate spellings and acoustic distances from* ACOUSTICRANK *detailed in Section 3.4.*

more accurate pronunciation. The method requires grapheme- or WordPiece-based ASR and TTS models, plus natural spoken examples of the words for which improved spellings are required. These resources are all fairly straightforward to construct, and none of them require a phonemic pronunciation dictionary. Our method is distinct from spelling reform efforts [12, 13], as it is not intended to generate *simplified* versions of original spellings for human use, per se. In fact, our method makes no use of the original spelling of a word, but generates a new spelling based solely on a spoken example; this typically results in a different spelling (e.g., the word originally spelled "Marseilles" is spelled "marssayy" by our method). Our method is applicable to any existing TTS grapheme-based system and does not require any changes or further training of that system.

### 1.3. Optional Human-in-the-Loop (HitL)

Our approach can optionally incorporate human intervention in the selection of the spelling that results in the best pronunciation. The human only needs to listen to a number of synthetic candidates, and pick the best; any native speaker of the language would be able to do this.

## 2. Spell4TTS

Spell4TTS is our proposed method for automatically deriving word spellings from spoken examples. The three stages are candidate spelling generation, synthesis, and ranking, as illustrated in Figure 1. In this paper, we provide concrete solutions to each stage, and compare a few different ways of performing the ranking. But the overall method does not depend on these details and alternative approaches could easily be used in any of the stages.

### 2.1. Stage 1: Generate candidate spellings

This stage needs to propose a set of spellings, amongst which there is at least one spelling that, when synthesised, will closely match the spoken ground-truth example. One option might be to propose all possible spellings, but that would be impractical, and most of them would be useless. Another option would be to perturb the original spelling by some means, creating many variants, but the means of doing this are not obvious, and this approach risks not finding a good spelling that is very different from the original. We only want spellings that are *acoustically-related* to the spoken example, so we use a simple ASR system. We note while a simple ASR system cannot guarantee that we generate an optimal spelling, it is sufficient to demonstrate the

efficacy of our method.

### 2.2. Stage 2: Synthesise candidates

All candidates from Stage 1 are synthesised using the particular TTS system for which we wish to improve pronunciation.

### 2.3. Stage 3: Rank candidates acoustically

The acoustic distance between each candidate and the spoken ground-truth example is measured, and candidates are selected. In the fully-automated version of our method, the top-ranked candidate is chosen. If there is an optional Human-in-the-Loop, they will choose amongst the top few candidates.

## 3. Experiments

Given the many design choices possible for the three stages, we limit the current work to testing three hypotheses:

H1– calculating acoustic distances using representations from self-supervised learning (SSL) will identify better-pronounced synthetic speech than when using MFCCs, because SSL representations are able to separate phonemic information from channel, speaker, and other unwanted properties. Testing this hypothesis first allows us to employ the best acoustic distance measure when testing subsequent hypotheses.

H2– the proposed method will find a spelling that results in a more accurate pronunciation than either the original spellings or the 1-best spelling from the ASR baseline (described shortly), because the method uses acoustic information to choose the spelling. This is the main claim of the current work.

H3– placing a Human-in-the-Loop in Stage 3 will find even better-sounding spellings than the fully-automatic method. This is a secondary claim.

### 3.1. Speech dataset & word-aligned spoken examples

We use the LJSpeech dataset to train our models. However, unlike the conventional TTS use case of the dataset, we partitioned it into two equally sized halves: $\mathcal{D}_{\text{train}}$ and $\mathcal{D}_{\text{test}}$. The former was employed for training the ASR and TTS models, while the latter was used to obtain a sizeable number of mispronounced out-of-vocabulary wordtypes. $\mathcal{D}_{\text{test}}$ was formed by taking utterances that contain the lowest frequency wordtypes one-by-one until it consisted of half of LJSpeech; the remainder were placed in $\mathcal{D}_{\text{train}}$. Consequently, $\mathcal{D}_{\text{test}}$ contains 13811 word types, out of which 8343 were absent from $\mathcal{D}_{\text{train}}$. $\mathcal{D}_{\text{train}}$ contains 5657 word types. Transcripts were normalised to contain only lowercase

alphabetic characters without punctuation. To acquire spoken examples for each word type in $\mathcal{D}_{\text{test}}$, we used word alignments from the Montreal Forced Aligner[1]. It is worth noting that although we adopted this high-resource word alignment solution, our method is adaptable to use a lower-resource approach or simply a purpose-recorded isolated word speech corpus.

## 3.2. Models

- Automatic Speech Recognition (ASR): We utilised a CTC end-to-end ASR architecture [14] to generate candidate spellings from spoken examples. The architecture comprised 2 CNN blocks with 128 and 256 channels, a 3 by 3 kernel size, and no time pooling or subsampling factor. These were followed by two Bidirectional LSTM layers with 512 hidden units and then two DNN layers, with 512 hidden units and 28 output units (26 alphabet characters, whitespace, and the blank token). We generated candidate spellings using CTC beam search decoding with a 1000 n-best list, beam size of 2000, and beam threshold of 50. To prevent generation of the whitespace token we manually set its probability to zero for all frame timesteps. In this work, we refrained from using an encoder-decoder ASR architecture with top-k or nucleus sampling, or language model rescoring, since these techniques might overpower the acoustic model, making it less likely to generate novel spellings. We used the SpeechBrain toolkit[2] to implement the model, and trained it on 90% of $\mathcal{D}_{\text{train}}$, with 10% split between validation and testing.

- Text-to-speech (TTS): We used a grapheme-based FastPitch TTS model to synthesise candidate spellings, adopting the same architecture as the original work [2]. We trained the FastPitch model for 1000 epochs using a batch size of 16 on $\mathcal{D}_{\text{train}}$, with monotonic alignment search [15] to obviate the need for external alignments. For waveform generation, we employed HifiGAN [16].

- Self-supervised speech representations (SSL): We investigated calculating acoustic distances between SSL speech representations derived from a HuBERT model[3] [8] implemented by the authors of [11]. This implementation can optionally extract 'soft' speech representations, which were claimed to better capture the nuances of pronunciations for voice conversion purposes.

## 3.3. Experiment 1 (H1)

Experiment 1 aimed to test H1 and determine the acoustic distance measure to use in Experiment 2. To design an effective measure it should compare two renditions of a word and be invariant to non-pronunciation variations in speech. To account for differences in sequence length, dynamic programming alignments, Dynamic Time Warping (DTW) or Levenshtein, are used with a local distance measure (cosine or Euclidean). Five types of speech features were extracted from synthesised candidates obtained in Stage 2, including SSL speech representations from layer 7 of the HuBERT model (which perform well on phone discrimination tasks [8, 17, 18]).

- MFCC (Euclidean+DTW): MFCCs are a traditional feature type commonly used in ASR systems and are engineered to

[1]https://montreal-forced-aligner.readthedocs.io
[2]https://speechbrain.github.io
[3]https://huggingface.co/facebook/hubert-base-ls960

remove speech information such as pitch. We extracted the first 12 MFCCs using the Librosa package[4].

- HUBERT-RAW (Cosine+DTW): Raw 768-dimensional HuBERT representations are the simplest to extract and have been shown to contain linguistic, prosodic, and semantic information in a form that is more easily linearly separable than traditional features like MFCCs [9, 10].

- HUBERT-CENTROID (Cosine+DTW): We extracted the centroids, i.e., the mean of vectors belonging to a cluster, using a 100 cluster k-means model trained on raw HuBERT representations extracted from LibriSpeech-960 [19]. Cluster centroids tend to have less information from paralinguistic aspects such as speaker ID, while still retaining information related to more subtle aspects, such as place or manner of articulation [20].

- HUBERT-CODE (Levenshtein): We also use the k-means model's discrete cluster IDs as features as they most aggressively discard paralinguistic information [21, 22]. This makes them more robust to differences unrelated to pronunciation.

- HUBERT-SOFT (Cosine+DTW): We use soft HuBERT features as a possible improvement over cluster IDs which can sometimes discard linguistic content, causing mispronunciations in voice conversion [11] and TTS [7]. These soft 256-dimensional representations were trained by predicting a distribution over the cluster IDs.

## 3.4. Experiment 2 (H2 & H3)

To investigate H2 and H3 we include three conditions which compare our proposed method SPELL4TTS with an ASR baseline and the original spellings, and we also incorporate Human-in-the-Loop refinement with the proposed method and the ASR baseline forming two additional conditions. Each of these five conditions devises a single spelling for each spoken example. These spellings are then synthesised and presented to the subjects in the listening test.

- ORIGINALSPELLING: The original unmodified spelling.

- ASR: The top-ranked (i.e. 1-best) spelling from the ASR model's hypotheses.

- ASR+HITL: We show the top $m$ highest likelihood spellings from the ASR hypotheses to a Human-in-the-Loop (HITL) who then chooses the one they judge best matches the spoken example in terms of pronunciation. For our experiments we set $m$ to 5. Although $m$ could be increased given time and budget, it is worthy to note that large values would be difficult for a human to filter effectively.

- ACOUSTICRANK: This condition uses our proposed method Spell4TTS to retrieve spellings in a fully automated fashion. We calculate acoustic distances using raw HuBERT features due to their superiority in Experiment 1 as discussed in Section 4.1.

- ACOUSTICRANK+HITL: Same as ACOUSTICRANK but we instead acoustically retrieve the top $m$ candidates which are then refined down to a single one by a Human-in-the-Loop in a similar fashion to ASR+HITL.

## 3.5. Evaluation & statistical analysis

We evaluate the pairs of conditions in Experiment 1 and 2 using subjective AB listening tests. We do not employ objective metrics as they may neglect more nuanced aspects of pronunciation,

[4]https://librosa.org

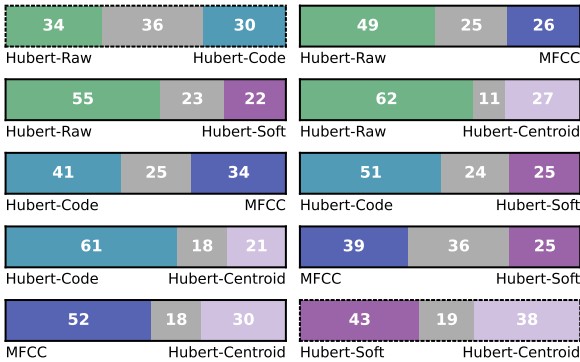

(a) *Proportion plots for the pairs of conditions shown to listeners. The middle grey region reflects 'no preference'. Note: the condition pairs* HUBERT-CODE *vs.* HUBERT-RAW *(p-value = 0.18) and* HUBERT-CENTROID *vs.* HUBERT-SOFT *(p-value = 0.2) are not statistically different from each other, and are highlighted with dotted line edges. All other pairs are statistically different.*

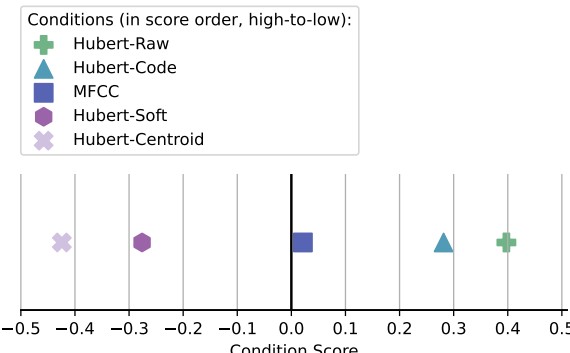

(b) *Condition scores estimated from a Bradley-Terry model. These can be regarded as the quality or strength of each condition (higher is better). Score values are 0.4 for* HUBERT-RAW, *0.28 for* HUBERT-CODE *0.02 for* MFCC, *-0.28 for* HUBERT-SOFT, *and -0.42 for* HUBERT-CENTROID.

Figure 2: *Experiment 1 listening test results, comparing how different acoustic features perform within our proposed method.*

such as syllable stress and coarticulation.

### 3.5.1. Listening test stimuli

We selected 100 orthographically opaque wordtypes for Experiment 1 and another 100 for Experiment 2 in the following way: First, to identify the wordtypes in $\mathcal{D}_{\text{test}}$ that would most likely be mispronounced by our TTS model we trained a weak G2P model[23] on $\mathcal{D}_{\text{train}}$ to calculate the phone-error-rate (PER) of each wordtype in $\mathcal{D}_{\text{test}}$ (according to ground-truth pronunciations in CMUDict [24]), and then select the 200 highest PER wordtypes over 7 characters long. Spoken audio for these wordtypes is then retrieved from $\mathcal{D}_{\text{test}}$ and is used by all the conditions except ORIGINALSPELLING to find spellings which are then synthesised by our TTS model. A selection of stimuli can be found on our samples page[5].

### 3.5.2. Listening test design

For each experiment we generated 100 stimuli from each of the 5 conditions. We then paired the conditions together making 10 pairs of conditions which altogether formed 1000 AB questions. Since 1000 questions were too many for any listener to rate in a single session, we broke them down into 10 subtests. Each subtest contained 100 AB questions and we used a Latin square to ensure that each subtest contained 10 AB questions from each pair of conditions. For each AB question participants listened to the spoken example of a wordtype, and two synthesised spellings. Both the question order and order of A and B were randomised on a per participant basis. They were prompted to select the synthesised rendition that most closely matches the spoken example in terms of pronunciation. They were also allowed to select a third 'no preference' option. We recruited 30 native English participants from North America using prolific[6].

### 3.5.3. Statistical testing

To determine significance of our listening test results we performed pairwise proportion Z-tests to determine whether two proportions are different according to their sample sizes. In our case for each pair of conditions we obtained 10 observations from each of the 30 participants, giving 300 in total.

We applied the Bradley-Terry model [25] to obtain scores that reflect the relative strength of each condition based on pairwise comparison data. Scores $s_i$ are estimated using the simultaneous equations induced by each pair of conditions $i$ and $j$, defined by Equation 1. Probabilities of pairwise comparisons are estimated using a wins count matrix $W_{i,j}$. We adjusted for the 'no preference' option by assigning half a 'win' to each condition. We estimated the Bradley-Terry parameters from $W_{i,j}$ using the Iterative Luce Spectral Ranking algorithm[7].

$$P(i > j) = \frac{s_i}{s_i + s_j} \tag{1}$$

# 4. Results

## 4.1. H1 is supported: HuBERT features outperform MFCCs for finding pronunciations via acoustic ranking

Figures 2(a) and 2(b) demonstrate that HuBERT features outperform MFCCs overall. Specifically, HUBERT-RAW performs the best, followed closely by HUBERT-CODE. This finding is interesting as these two conditions are on opposite sides of the spectrum in terms of discarding paralinguistic information. It is worth noting that they are not significantly different from each other according to a pairwise proportion Z-test (p-value = 0.18). In contrast, MFCC features perform reasonably well, outperforming HUBERT-SOFT and HUBERT-CENTROID. Overall our findings suggest that HuBERT features should be used to maximize performance, but MFCCs can be adopted if a simpler pipeline is desired.

---

[5]https://spell4tts.github.io/samples
[6]https://www.prolific.co

[7]http://choix.lum.li

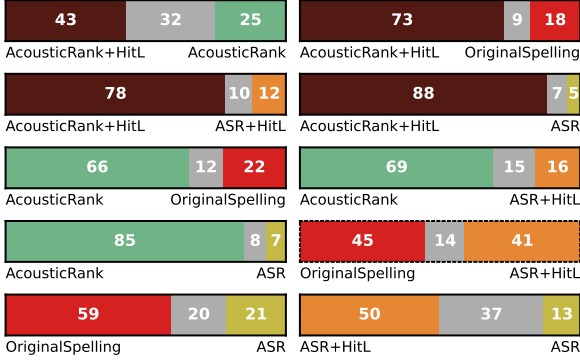

(a) *Proportion plots. Note: the condition pair* ASR+HITL *vs.* ORIGINALSPELLING *is not statistically different from each other (p-value = 0.35), and is highlighted with a dotted line edge. All other pairs are statistically different.*

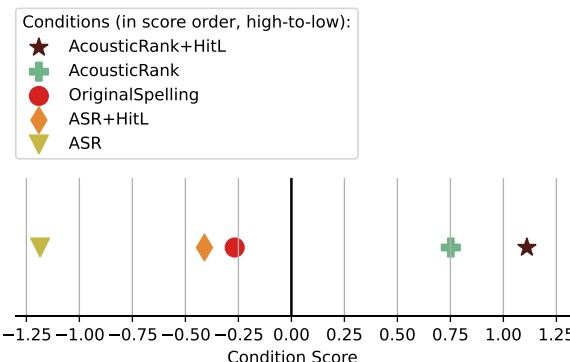

(b) *Bradley-Terry model condition scores. Score values are 1.1 for* ACOUSTICRANK+HITL, *0.75 for* ACOUSTICRANK, *-0.27 for* ORIGINALSPELLING, *-0.41 for* ASR+HITL, *and -1.2 for* ASR. *Note that for convenience* ACOUSTICRANK *is assigned the same symbol and colour as the condition* HUBERT-RAW *in Figure 2 as it retrieves pronunciations in an identical way.*

Figure 3: *Experiment 2 listening test results, comparing our proposed method with ASR baselines and original spellings.*

### 4.2. H2 is supported: Acoustic ranking outperforms ASR baselines and original spellings

Figures 3(a) and 3(b) both show that the proposed fully automated method, ACOUSTICRANK, is shown to be significantly better than both ASR baselines, indicating that more optimal pronunciations often lie outside the ASR 1-best or 5-best lists, and that we can retrieve them automatically. Interestingly, the ASR baselines were found to be no better than using original spellings, even when incorporating Human-in-the-Loop assistance. Notably, ASR performed particularly poorly in comparison to original spellings.

### 4.3. H3 is supported: Incorporating a Human-in-the-Loop finds better spellings

Furthermore from Figures 3(a) and 3(b) we observe that incorporating Human-in-the-Loop assistance consistently led to the discovery of better pronunciations and further enhanced the performance of our proposed method. Section 4.4.2 delves deeper into this result.

### 4.4. Further analysis

#### 4.4.1. Automated retrieval statistics

Figure 4 reveals a roughly even distribution of ASR n-best ranks from 1 to 1000. This evidence, along with the findings presented in Section 4.3, suggests that optimal spellings are often located far beyond the scope of small ASR n-best lists. As a result, a manual search for optimal spellings is infeasible, thereby strengthening the rationale for using our proposed method.

#### 4.4.2. Human-in-the-loop refinement statistics

Figures 5 and 6 demonstrate that the optimal spellings selected by our Human-in-the-Loop approach are approximately evenly distributed among the top-5 spellings generated for both ASR+HITL and ACOUSTICRANK+HITL. Notably, the superior performance of ACOUSTICRANK+HITL over ACOUSTICRANK and Figure 6 both suggest that our proposed acoustic ranking method is effective at reducing a large ASR n-best list to a shortlist of potentially optimal spellings. However, our

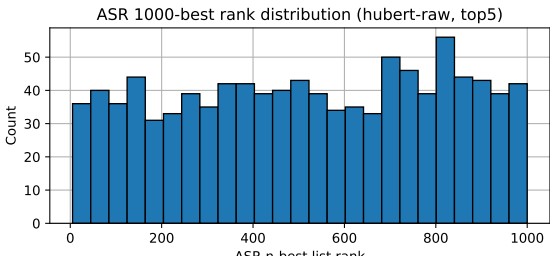

Figure 4: *Histogram of ASR n-best ranks of the top-5 spellings retrieved using* HUBERT-RAW/ACOUSTICRANK *for the 200 evaluated wordtypes from Experiments 1 and 2.*

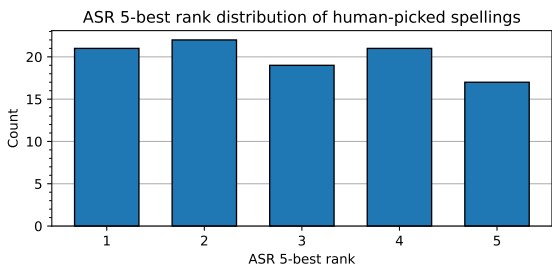

Figure 5: *Distribution of ASR n-best ranks of the human selected spellings in* ASR+HITL.

method did not consistently rank the human-selected spelling at the very top. This could be due to a failure to detect small differences in pronunciation. It could also be related to our observation that within the top-5 spellings retrieved for ACOUSTICRANK+HITL, sometimes all of the spellings have different and slight mispronunciations, requiring the Human-in-the-Loop to select one of these spellings somewhat arbitrarily. We believe that further research to seek an improved acoustic distance measure might alleviate this issue.

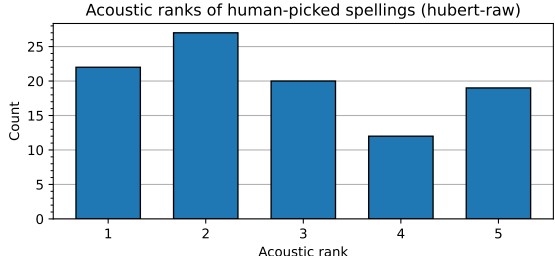

Figure 6: *Distribution of acoustic ranks of the human selected spellings in* ACOUSTICRANK+HITL.

## 5. Conclusion

This paper introduces a novel cost-effective method for improving the pronunciation of any existing text-to-speech (TTS) system, without requiring additional model training. The proposed method, Spell4TTS, automatically generates candidate spellings and then filters them via acoustic ranking. The resulting spellings are both human-interpretable and editable, making the method suitable for deployment in production environments. We believe that this method will be particularly beneficial for low-resource TTS. Our experiments demonstrate that the method devises pronunciations that outperform those obtained using an automatic speech recognition (ASR) baseline and the original spellings. Furthermore, the method can be augmented with human judgements to further enhance the quality of the pronunciations. We also show that ranking candidate spellings in the acoustic space of self-supervised speech representations, as opposed to traditional hand-engineered features, can yield further improvements in pronunciation. In future work, we plan to adapt Spell4TTS to mine pronunciations from multi-speaker corpora, incorporate stress markers and syllable boundaries, explore its utility in transcribing region-to-region pronunciation variation, and apply it to a low-resource language in a real-world scenario.

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
