# OpenReview forum: "Spell4TTS: Acoustically-informed spellings for improving text-to-speech pronunciations"
_Interspeech.org/2023/Workshop/SSW — SSW12_

### Official Review · Reviewer_oicy · 2023-06-02
**A good method of finding better pronunciations for grapheme based TTS**

**Rating:** 7
**Confidence:** 4

**Review:**

Key Strength of the paper:
Finding better pronunciations for grapheme based TTS, where the output pronunciation cannot be controlled as with phoneme based TTS.

Main Weakness of the paper:
This method aims to find the sequence of graphemes that corresponds to the best pronunciation. However, since the TTS model is not fine-tuned and it operates on the grapheme-level, this approach must be applied to all of the words in a specific language in order to obtain the optimized pronunciations. There can be many words, such as names, addresses that cannot be known beforehand and perhaps in these cases the model will perform poorly.

Novelty/Originality:
Adequately novel, regarding the three stage approach for finding the appropriate grapheme sequences and utilization of state-of-the-art audio representations.

Technical Correctness:
Correct in the context of comparisons between the suggested methods.

Suggestions for improvement:
Perhaps a comparison with a similar phoneme-based TTS model would be good, because it would show if the phoneme-based model provides an upper bound in pronunciation modeling, or if the proposed method has the potential to show better or equal results.

Quality of References:
Adequate references.

Clarity of Presentation:
Clear presentation and good usage of English.

---

> ### Author Response · Authors · 2023-06-26
> **Response to finetuning.**
>
> Thank you for your review oicy, I very much appreciate your comments.
>
> With regard to the main weaknesses:
>
> - We believe that finetuning the TTS model (or even the ASR one) is a potential future extension of our method. As it stands with no finetuning, our main target use-case is a TTS system trained with grapheme-inputs only, where our method can be used to discover spelling-based pronunciations if and when needed to correct mispronounced words. We believe this approach need not be applied to all words in a specific language since the implicit grapheme-to-pronunciation model within a grapheme-input TTS system will likely pronounce in-training wordtypes correctly, and may pronounce a large proportion of out-of-training wordtypes correctly too. We agree that there are many such loan words or proper nouns such as names and addresses that cannot be known beforehand, we believe our method can be applied to correct these words if access to a spoken example is available.
>
> Suggestions for improvement:
>
> - We leave comparison with phoneme-based TTS to future work due to lack of time to train and incorporate a phoneme-based TTS system.
>
> Best

---

### Official Review · Reviewer_yaax · 2023-06-02
**novel idea with clear results for improved TTS pronunciations**

**Rating:** 7
**Confidence:** 4

**Review:**

Key Strength of the paper
- an extension for any TTS system which can help better pronunciations
- in general, pronunciations are improved, but human-in-the-loop might be still necessary

Main Weakness of the paper
- not clear how easy it is to use for other languages
- it was only tested on LJSpeech, i.e. single English speaker

Novelty/Originality, taking into account the relevance of the work for the SSW audience
- the idea is clearly novel

Technical Correctness, is the work technically and/or scientifically solid? Are sufficient details provided to allow any experiments to be reproduced or equivalent experiments run?
- the details of the methods and experiments are clear

Suggestions for improvement
- Sec 2.1: how it is guaranteed that at least one spelling will closely match the spoken ground-truth example?
- Sec 3.1: it is not clear how MFA and separate ASR were used
- Sec 4.1: what about the significances between the other versions?


Quality of References, is it a good mix of older and newer papers? Do the authors show a good grasp of the current state of the literature? Do they also cite other papers apart from their own work?
- refs are OK

Clarity of Presentation, the English does not need to be flawless, but the text should be understandable
- the main text is more than 5 pages, but it is OK in case of SSW
- the sound samples at https://spell4tts.github.io support the results

---

> ### Author Response · Authors · 2023-06-26
> **Addressing multilinguality and using multi-speaker spoken examples.**
>
> Thank you for your review yaax, I very much appreciate your comments.
>
> With regard to the main weaknesses:
>
> - Multilinguality: multilinguality is, of course, a promising application of this method, however, we didn’t have time to test our method with additional languages. Future work should look to address this gap in knowledge. Observing how well this method will work with languages with smaller alphabets is of scientific interest. We believe the method is still useful for high resourced, orthographically opaque languages such as English as smaller TTS teams may not have access to phonemic resources for training or have phonemically trained practitioners when addressing pronunciation errors of their phoneme-trained TTS systems.
> - Multispeaker: being able to generate candidate spellings from non-target speakers is another important functionality which we were unable to test due to limitations in time. We believe that if the ASR system is trained with sufficient number of speakers, then it would likely become more useful for our task of candidate spelling generation.
>
> Suggestions for improvement:
>
> - We will state that our current method has no guarantee for improvement.
> - We added a little more detail regarding use of MFA.
>
> Best

---

### Decision · Program_Chairs · 2023-06-14

**Decision:**

Accept

**Comment:**

SSW2003 received 45 papers. The acceptance rate is 82%. We are pleased to inform you that your paper has been accepted by the SSW2023 Program Committee. Please read the reviews carefully and submit your camera-ready paper by June 28th. Most reviewers performed a detailed review. Please answer to their questions and consider their comments. Note that camera-ready papers are credited with one extra page to allow authors to consider reviewers’ suggestions. So max 7 pages in total including figures & refs.
The deadline for submitting the revised version (with full non-anonymized authors and refs!) is 28th June.